# Multivariate Time Series Forecasting with Fourier Neural Filter

## Abstract

Multivariate time series forecasting has been suffering from the challenge of capturing both temporal dependencies within variables and spatial correlations across variables simultaneously. Current approaches predominantly repurpose backbones from Natural Language Processing (NLP) or Computer Vision (CV) (*e.g.*, Transformers), which fail to adequately address the unique properties of time series (*e.g.*, periodicity and fluctuation). The research community lacks dedicated backbones incorporating temporal-specific inductive biases, depending on domain-agnostic backbones supplemented with auxiliary techniques (*e.g.*, signal decomposition). We introduce Fourier Neural Filter (FNF) as the backbone and Dual Branch Decoupler (DBD) as the architecture to provide exceptional learning capabilities and optimal learning pathways for spatiotemporal modeling, respectively. Our theoretical analysis proves that FNF integrates time-domain and frequency-domain analysis while enabling adaptive truncation of noise components within a unified backbone that extends naturally to spatial modeling. Through the lens of information bottleneck theory, we reveal that DBD delivers superior gradient flow and representation capacity, enabling it to effectively capture local and global, temporal and spatial information comprehensively. Our empirical evaluation on 12 public benchmark datasets, encompassing both multivariate long-term and short-term forecasting tasks, demonstrates state-of-the-art performance compared to existing advanced baseline models. Notably, our approach achieves these results without any auxiliary techniques, suggesting that properly designed neural architectures can capture the inherent properties of time series, potentially transforming time series modeling in scientific and industrial applications.

## 1 Introduction

Time series forecasting, which estimates future values based on historical values, has attracted substantial academic attention and found widespread application across diverse domains, including energy Zhang et al. (2025), meteorology Pathak et al. (2022), and transportation Li et al. (2018). Recent research has increasingly focused on multivariate long-term forecasting Zhou et al. (2021), which presents several significant challenges. Extended forecast horizons

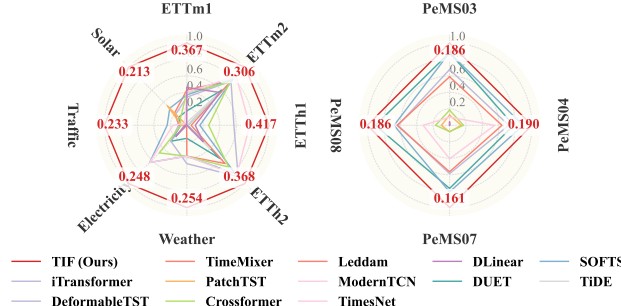

Figure 1: **Radar chart of model performance across 12 benchmark datasets encompassing multivariate long-term and short-term time series forecasting.** The chart displays the average MAE across different forecast horizons. Our proposed TiF consistently outperforms other strong baseline models.

inevitably increase uncertainty and degrade prediction accuracy, while complex temporal dependencies within variables and spatial correlations across variables further complicate modeling, particularly with high-dimensional variables. Consequently, developing neural architectures capable of simultaneously capturing heterogeneous temporal and spatial patterns has become critical for advancing time series forecasting.

For **within-variable** temporal modeling, the research community has developed several elaborate approaches. These include: (i) decomposing time series into trend, seasonal, and residual terms Oreshkin et al. (2020); Challu et al. (2023); (ii) deconstructing time series into frequency components for *multi-resolution modeling* Dai et al. (2024a); and (iii) segmenting sequences into patches of uniform Nie et al. (2023); Lee et al. (2023) or varying sizes Chen et al. (2024b); Yu et al. (2024b) to capture both local and global temporal dependencies through *multi-scale modeling*. More sophisticated techniques integrate multi-resolution and multi-scale modeling simultaneously Wang et al. (2025), establishing comprehensive frameworks for time series representation learning.

Despite these advances, existing approaches primarily rely on backbones borrowed from Natural Language Processing (NLP) or Computer Vision (CV), such as Transformers Vaswani et al. (2017), CNNs Wu et al. (2023); Luo & Wang (2024b), and MLPs Zeng et al. (2023). These domain-agnostic backbones cannot fully address the inherent properties of time series without auxiliary techniques (*e.g.*, signal decomposition). To address this fundamental limitation, we propose the **Fourier Neural Filter (FNF)**, a novel nonlinear integral kernel operator that integrates temporal-specific inductive biases directly into the backbone design. Mathematically, FNF extends the standard Fourier Neural Operator (FNO) Li et al. (2021) by introducing an input-dependent kernel function that enables selective activation of time-domain and frequency-domain information through Hadamard product operations, making it particularly effective for capturing the unique properties of time series. On the other hand, we incorporate adaptive truncation following complex operation, making it capable of effectively suppressing the noise components. Additionally, FNF offers several key advantages: (i) it naturally extends to spatial modeling (explained in Section 3.1.1); (ii) it achieves $O(N \log N)$ computational complexity compared to the $O(N^2)$ complexity of Transformers; and (iii) in contrast to other purely Fourier-based models Zhou et al. (2022); Yi et al. (2023); Xu et al. (2023); Yu et al. (2024a), it allows for dynamic information flow modulation between time-domain and frequency-domain, constructing a unified time-frequency representation space.

For **cross-variable** spatial modeling, the researchers have developed diverse techniques: (i) *independent variable modeling* Zeng et al. (2023); Nie et al. (2023), which maintains stability but ignores inter-variable interactions; (ii) *unified variable modeling* Zhang & Yan (2023); Liu et al. (2023a), which comprehensively captures relationships but exhibits sensitivity to irrelevant variable disturbances; and (iii) *hierarchical variable modeling* Chen et al. (2024a), which provides a compromise approach but constrains flexibility by confining relationship patterns within predetermined cluster boundaries. These techniques highlight the fundamental trade-offs in spatial modeling and emphasize the need for adaptive systems that can effectively balance these conflicting demands.

To address above challenges, we propose the **Dual Branch Decoupler (DBD)**. From an information bottleneck perspective Tishby & Zaslavsky (2015), this parallel dual-branch architecture is able to optimize information extraction and compression in multivariate time series by maintaining separate processing pathways for temporal and spatial patterns. Unlike unified techniques that suffer from the curse of dimensionality or sequential techniques Zhang & Yan (2023); Chen et al. (2024b) that experience cascading information loss due to unequal information processing, DBD ensures that each branch independently achieves optimal trade-offs between information extraction and compression while providing short and direct gradient flow.

The two proposed techniques FNF and DBD collectively form the innovative model we aim to introduce, specifically designed for time series: **Time Filter (TiF)**. Benefiting from the completely isolated information flow of DBD, TiF is able to effectively capture local and global, temporal and spatial information, more comprehensive than other baseline models.

To validate our proposed model, we conduct comprehensive experiments on 12 public benchmark datasets, encompassing both multivariate long-term and short-term forecasting tasks. Our extensive evaluation demonstrates that our approach achieves state-of-the-art results compared to existing advanced baseline models, as shown in Fig. 1. Additionally, we conduct fair comparisons with three other purely Fourier-based models to further demonstrate the effectiveness of our approach.

Our contributions are as follows: (1) We propose FNF, a unified backbone integrating time-domain and frequency-domain analysis, specifically designed for time series modeling; (2) We introduce DBD, and prove its effectiveness for spatiotemporal modeling of time series both theoretically and empirically; (3) The proposed model TiF achieves state-of-the-art performance in both multivariate long-term and short-term time series forecasting.

## 2 RELATED WORK

**Distribution Shift**   The statistical properties of time series, such as mean and variance, tend to change over time, creating fundamental challenges for time series modeling Passalis et al. (2019); Liu et al. (2025). Researchers have developed various solutions to address this issue. RevIN Kim et al. (2021) applies instance normalization on input sequences and performs de-normalization on output sequences. Dish-TS Fan et al. (2023) extracts distribution coefficients for both intra- and inter-variable to mitigate distribution shift. SAN Liu et al. (2023b) addresses non-stationarity by using a lightweight neural network to model evolving local statistical properties. FAN Ye et al. (2024) handles dynamic trend and seasonal patterns by employing Fourier transform to identify key frequency components. DDN Dai et al. (2024b) eliminates non-stationarity by applying wavelet transform to implement normalization of time and frequency domains within the sliding window. Notably, Li *et al.* Li et al. (2023) demonstrated that after using instance normalization, excellent results can be easily obtained with just a simple linear layer. Our work employs basic instance normalization while introducing the FNF backbone to strengthen the ability to process the inherent non-stationarity of time series.

**Patch Embedding**   Time series patching has evolved from simple segmentation to elaborate strategies that balance local and global information extraction Nie et al. (2023); Lee et al. (2023). These techniques include overlapping Luo & Wang (2024b) or non-overlapping Zhang & Yan (2023) patching, variable length patching Yu et al. (2024b), and hierarchical patching Huang et al. (2024). Recent research has explored adaptive patching strategies based on input properties of time series Chen et al. (2024b). Although these approaches enhance representation learning capability , they are still limited to enhancing existing backbones through *preprocessing* without addressing fundamental design principles. In contrast, our work maintains basic non-overlapping patching for fair experimental comparison while introducing the FNF backbone specifically designed to unify local time-domain and global frequency-domain processing for time series.

**Non-Autoregressive Decoding**   To alleviate the error accumulation problem in autoregressive decoding, non-autoregressive approaches Wu et al. (2021); Zhou et al. (2021); Wu et al. (2023); Liu et al. (2023a) have become the standard paradigm for time series forecasting. This technique simultaneously generates all future values through a linear layer, rather than recursively using previous predictions as inputs to obtain future values. While patch-based autoregressive models excel in large-scale time series foundation models Das et al. (2024); Shi et al. (2025), non-autoregressive approaches perform better in typical forecasting tasks. Recent research has identified that non-autoregressive approaches implicitly assume conditional independence between future values, ignoring the autocorrelation inherent in time series Wang et al. (2024a). Our proposed FNF and DBD enhance representation learning capability within this established system, despite the theoretical limitations of the non-autoregressive paradigm.

## 3 METHODOLOGY

In this section, we establish the theoretical foundations of our proposed Fourier Neural Filter (FNF) backbone and Dual Branch Decoupler (DBD) architecture.

### 3.1 FOURIER NEURAL FILTER (FNF)

While FNO Li et al. (2021) has demonstrated remarkable effectiveness in modeling complex dynamic systems and solving partial differential equations through fixed integral kernel, our proposed FNF (Fig. 2) makes a critical leap forward: introducing an input-dependent integral kernel that can allow for adaptive and dy-

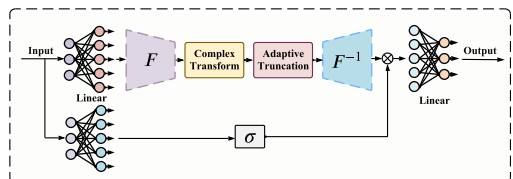

Figure 2: **Schematic diagram of our proposed Fourier Neural Filter (FNF) backbone.**

namic information flow between time-domain and frequency-domain, constructing a unified time-frequency representation space. Intuitively, if FNO applies a fixed lens to process all input signals, then FNF continuously adjusts the lens based on the preceding scene, achieving more detailed information extraction and more robust pattern recognition. We analyze the theoretical underpinnings

of FNF by examining integral kernel, global convolution, selective activation, complex transform, adaptive truncation and the connection to Transformer backbone.

### 3.1.1 INTEGRAL KERNEL

**Definition 1** FNO is defined via a fixed integral kernel operator:

$$(Kv)(x) = \int_D \kappa(x, y)v(y)\, dy, \tag{1}$$

where $\kappa : D \times D \to \mathbb{R}$ is the kernel function and $v : D \to \mathbb{R}$ is the input function. Through the Fourier transform, FNO can be formulated in the frequency domain as:

$$(Kv)(x) = \mathcal{F}^{-1}(R_\phi \cdot \mathcal{F}(v))(x), \tag{2}$$

where $R_\phi = \mathcal{F}(\kappa)$ denotes the parameterized frequency-domain kernel.

**Definition 2** FNF can be defined through an adaptive integral kernel operator:

$$(Kv)(x) = \int_D \kappa(x, y; v)v(y)\, dy, \tag{3}$$

where $\kappa(x, y; v)$ is the input-dependent kernel function. In implementation, FNF can also be formulated as:

$$(Kv)(x) = T(G(v) \odot P(v))(x), \tag{4}$$

$$P(v)(x) = \mathcal{F}^{-1}(R_\phi \cdot \mathcal{F}(H(v)))(x), \tag{5}$$

where $G(v)$, $H(v)$, and $T(v)$ denote the linear transform used for expansion or compression, and $\odot$ is the Hadamard product operation.

**Remark 1** The fundamental distinction between FNO and FNF lies in their kernel functions: FNO employs a fixed kernel $\kappa(x, y)$, whereas FNF applies an input-dependent kernel $\kappa(x, y; v)$, enabling adaptive information flow modulation between time-domain and frequency-domain, constructing a unified time-frequency representation space.

**Remark 2** Although FNF is originally motivated by spatial-temporal operator learning, its formulation is not restricted to continuous spatial or temporal domains. For multivariate time series, the variable dimension can be regarded as a discrete one-dimensional domain, where each variable index corresponds to a point in this domain. Under this view, applying FNF is equivalent to learning a global operator to approximate this cross-variable correlation, through Fourier modes as a universal basis. This perspective provides a theoretical justification: even in the absence of natural sequence or geometric structures, FNF can still serve as an efficient kernel approximation method on the variable index space, analogous to kernel methods in classical machine learning Cortes & Vapnik (1995); Seeger (2004). In this sense, FNF plays a role in cross-variable learning that is conceptually parallel to Transformers Gorishniy et al. (2021), where variable dimensions are treated as tokens, with the exception that FNF employs spectral convolution instead of attention mechanism.

### 3.1.2 GLOBAL CONVOLUTION

**Definition 3** When the kernel function $\kappa(x, y) = \kappa(x - y)$ exhibits translation invariance, the fixed integral kernel operator in FNO reduces to a global convolution Li et al. (2021):

$$(Kv)(x) = \int_D \kappa(x - y)v(y)\, dy = (\kappa * v)(x). \tag{6}$$

**Definition 4** Similarly, when the kernel function $\kappa(x, y; v) = \kappa(x - y; v)$ maintains translation invariance, the adaptive integral kernel operator in FNF becomes a gated global convolution:

$$(Kv)(x) = \int_D \tilde{\kappa}(x - y; v)v(y)\, dy = (\tilde{\kappa}(\cdot; v) * v)(x). \tag{7}$$

**Remark 3** Translation invariance enables efficient computation of integral operator through Fourier transform in both FNO and FNF. Beyond this shared efficiency, the gated global convolution in FNF significantly enhances representation capacity by employing an input-dependent kernel $\tilde{\kappa}(\cdot; v)$, which adaptively modulates filtering behavior while preserving computational efficiency.

### 3.1.3 SELECTIVE ACTIVATION

**Definition 5** The selective activation operates an element-wise multiplication in the time domain; in the frequency domain, this operation is mathematically equivalent to the convolution operation between $G(v)(x)$ and $P(v)(x)$:

$$\mathcal{F}(G(v) \odot P(v))(\omega) = (\hat{G}(v) * \hat{P}(v))(\omega). \tag{8}$$

This formula can be viewed as approximate magnitude modulation and phase addition when the signal $G(v)$ is relatively smooth or narrow:

$$(G(v) \odot P(v))_i \approx |G(v)_i| \cdot |P(v)_i| \cdot e^{i(\theta_{G(v)_i} + \theta_{P(v)_i})}, \tag{9}$$

where $|G(v)_i|$ and $|P(v)i|$ represent magnitudes and $\theta G(v)i$ and $\theta P(v)_i$ represent phases.

**Remark 4** This formulation reveals how selective activation effectively achieves the joint time–frequency modulation: it enhances informative mid-/high-frequency components while suppressing redundant low-frequency ones on the magnitude side, and simultaneously provides flexible alignment on the phase side. This design alleviates the well-known over-smoothing effect and bandwidth bottleneck Rahaman et al. (2019) of FNO and improves the representation learning capability .

### 3.1.4 COMPLEX TRANSFORM

**Definition 6** The complex transform operates on the complex-valued input $z = z_r + iz_i$ with complex weights $W = W_r + iW_i$ and biases $b = b_r + ib_i$:

$$L(z) = (W_r z_r - W_i z_i + b_r) + i(W_r z_i + W_i z_r + b_i). \tag{10}$$

**Remark 5** To reduce the parameter count, we adopt the block-diagonal structure for the weights Guibas et al. (2022) and implement two complex transform layers equipped with the GELU activation function Hendrycks & Gimpel (2016).

### 3.1.5 ADAPTIVE TRUNCATION

**Definition 7** The adaptive truncation operates through a Softshrink function Donoho (2002) with a learnable threshold:

$$S_\lambda(z) = \begin{cases} (|z| - \lambda)\frac{z}{|z|}, & \text{if } |z| > \lambda \\ 0, & \text{if } |z| \leqslant \lambda, \end{cases} \tag{11}$$

where $\lambda$ is the threshold parameter.

**Remark 6** The Softshrink function preserves phase while adaptively sparsifying the frequency spectrum. Unlike traditional approaches with fixed thresholds, we make $\lambda$ a learnable parameter to automatically discover unimportant components in different frequency components. This adaptive technique enables our backbone to: (1) balance noise suppression and signal preservation without manual tuning, (2) adapt to varying input properties, and (3) achieve superior denoising performance through end-to-end optimization.

### 3.1.6 CONNECTION TO TRANSFORMERS

**Functionality** FNF represents a unified backbone that implements core Transformer functions through alternative computational mechanisms. The gated global convolution in FNF performs comprehensive information interaction across all positions, analogous to token mixing in Transformer. Furthermore, the linear transformations ($T(v)$, $G(v)$, and $H(v)$) in FNF can be expanded to replicate the functionality of Feed-Forward Network (FFN). FNF establishes functional equivalence between different backbones while maintaining different computational paths.

**Complexity** FNF achieves token mixing with $O(N \log N)$ computational complexity through Fourier transform, compared to the $O(N^2)$ complexity of standard Transformers for sequence length $N$. Moreover, FNF typically requires fewer parameters while maintaining comparable performance, making it particularly efficient for modeling spatiotemporal patterns in multivariate time series.

## 3.2 DUAL BRANCH DECOUPLER (DBD)

Through the lens of information bottleneck theory Tishby & Zaslavsky (2015), spatiotemporal modeling architectures can be categorized into three paradigms: unified (which suffers from the curse of dimensionality) Alemi et al. (2017), sequential (which creates information bottlenecks) Zhang & Yan (2023); Chen et al. (2024b), and parallel (which preserves information through independent branches) Qiu et al. (2025). Our DBD architecture adopts the parallel approach to maximize gradient flow and representation capacity.

**Unified** The unified paradigm captures temporal and spatial through a single operation $X \rightarrow T \rightarrow Y$, where $T$ must simultaneously encode both temporal and spatial information. This approach suffers from the curse of dimensionality and requires substantially more parameters to achieve comparable performance, while carrying the risk of overfitting.

**Sequential** The sequential paradigm implements cascaded information processing $X \rightarrow T_1 \rightarrow T_2 \rightarrow Y$, where temporal processing precedes spatial processing. This architecture inherently suffers from information loss due to the information processing inequality: $I(X;Y) \geqslant I(T_1;Y) \geqslant I(T_2;Y)$, where each stage acts as an information bottleneck.

**Parallel** The parallel paradigm maintains independent information processing branches with direct access to the original input Wang et al. (2018):

$$X \rightarrow \left\{ \begin{array}{c} T_1 \\ T_2 \end{array} \right. \rightarrow T \rightarrow Y . \tag{12}$$

This approach ensures each branch independently achieves optimal information compression, with each optimizing its own objective function:

$$\min_{T_1} [I(T_1;X) - \beta_1 I(T_1;Y)]$$
$$\min_{T_2} [I(T_2;X) - \beta_2 I(T_2;Y)] , \tag{13}$$

where $\beta_1$ and $\beta_2$ control the trade-off between compression and performance for each branch.

**Gradient Flow** The parallel architecture offers significant advantages in gradient flow with independent branches maintaining short and direct gradient paths:

$$\frac{\partial L}{\partial \theta_1} = \frac{\partial L}{\partial g} \cdot \frac{\partial g}{\partial f_1} \cdot \frac{\partial f_1}{\partial \theta_1}$$
$$\frac{\partial L}{\partial \theta_2} = \frac{\partial L}{\partial g} \cdot \frac{\partial g}{\partial f_2} \cdot \frac{\partial f_2}{\partial \theta_2}. \tag{14}$$

These shorter gradient paths significantly reduce the risk of vanishing or exploding gradients, and enable efficient parallelization during both forward and backward passes, accelerating convergence without sacrificing model performance.

**Representation Capacity** The parallel architecture demonstrates superior representation capacity through complementary information retention. Due to the orthogonal nature of temporal and spatial information, their joint representation typically satisfies:

$$I(T_1, T_2;Y) > \max\{I(T_1;Y), I(T_2;Y)\} . \tag{15}$$

This inequality holds because temporal and spatial features capture fundamentally different aspects of the input: temporal features extract dynamic patterns across time, while spatial features encode structural relationships within each time point. Their combination provides a more comprehensive view than either branch could achieve alone.

## 4 MODEL

**Overview** We denote the input sequence as $X = (x_1, \ldots, x_T) \in \mathbb{R}^{M \times L}$ with $L$ lookback window and $M$ variables, and the output sequence $Y = (y_1, \ldots, y_T) \in \mathbb{R}^{M \times H}$ with $H$ forecast horizon. Our model TiF as illustrated in Fig. 3.

**Normalization**  To address distribution shift inherent in time series, we implement instance normalization Kim et al. (2021):

$$\hat{x}_{kt}^{(i)} = \frac{x_{kt}^{(i)} - \mathbb{E}\left[x_{kt}^{(i)}\right]}{\sqrt{\mathrm{Var}\left[x_{kt}^{(i)}\right] + \epsilon}}, \tag{16}$$

where $\mathbb{E}\left[x_{kt}^{(i)}\right] = \frac{1}{T_x}\sum_{j=1}^{T_x} x_{kj}^{(i)}$ and $\mathrm{Var}\left[x_{kt}^{(i)}\right] = \frac{1}{T_x}\sum_{j=1}^{T_x}\left(x_{kj}^{(i)} - \mathbb{E}_t\left[x_{kt}^{(i)}\right]\right)^2$ denote the mean and standard deviation vectors of the input sequence, with $\epsilon$ added to maintain numerical stability.

**Embedding**  Each variable of input sequence $X \in \mathbb{R}^{M \times L}$ is divided into non-overlapping patches $X_p' \in \mathbb{R}^{M \times N \times P}$, where each patch has a length of $P$ and the number of patches $N = \left\lceil \frac{L}{P} \right\rceil$, then each patch is mapped to a patch token $X_p \in \mathbb{R}^{M \times N \times D}$ through a linear layer Nie et al. (2023):

$$X_p' = \mathrm{Patching}(X), \quad X_p = \mathrm{Embedding}(X_p'), \tag{17}$$

where each patch can be viewed as local temporal dependencies within its respective time window $P$, which is particularly important for time series analysis, as it enables the capture of nuanced temporal patterns that may be obscured in global temporal modeling.

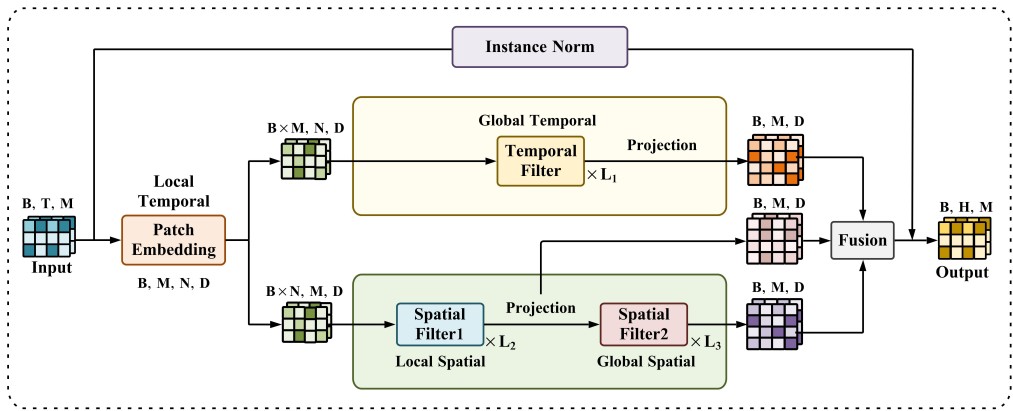

Figure 3: **Schematic diagram of our proposed Time Filter (TiF) architecture.** Our model integrates dedicated components, including Instance Normalization, Patch Embedding, Temporal Filter, Spatial Filter, and Fusion, with each tailored to a distinct facet of the time series forecasting challenge.

**Temporal Filter**  We apply FNF to the dimension of the number of patches $N$ to capture global temporal dependencies:

$$X_t' = \mathrm{FNF}_t(X_p). \tag{18}$$

The output embedding $X_t' \in \mathbb{R}^{M \times N \times D}$ is flattened, and projected through a linear layer to obtain $X_t \in \mathbb{R}^{M \times D'}$, which stands for the representation that aggregates local and global temporal dependencies:

$$X_t = \mathrm{Linear}(\mathrm{Flatten}(X_t')). \tag{19}$$

**Spatial Filter**  We apply FNF to the dimension of the number of variables $M$ to capture local spatial correlations:

$$X_{ls}' = \mathrm{FNF}_{ls}(X_p). \tag{20}$$

The output embedding $X_t' \in \mathbb{R}^{N \times M \times D}$ is transposed, flattened, and projected to obtain $X_{ls} \in \mathbb{R}^{M \times D'}$, which stands for the representation that aggregates local spatial correlations:

$$X_{ls} = \mathrm{Linear}(\mathrm{Flatten}(\mathrm{Transpose}(X_{ls}'))). \tag{21}$$

Subsequently, we similarly apply FNF to $X_{ls} \in \mathbb{R}^{M \times D'}$ for capturing global spatial correlations:

$$X_{gs} = \mathrm{FNF}_{gs}(X_{ls}). \tag{22}$$

**Fusion** The final step is to fuse the representations on behalf of the three different spatiotemporal patterns $X_t, X_{ls}, X_{gs} \in \mathbb{R}^{M \times D'}$, to obtain the output sequence $Y \in \mathbb{R}^{H \times M}$:

$$Y = \text{Linear}_3 \circ \text{LayerNorm} \circ \text{Linear}_2 \circ \text{GELU} \circ \text{Linear}_1 \circ \text{Concat}(X_t, X_{ls}, X_{gs}). \quad (23)$$

## 5 EXPERIMENT

In this section, to validate the effectiveness of our proposed TiF, we conduct extensive experiments on a variety of time series forecasting tasks, including both long-term and short-term forecasting.

**Baselines** For long-term forecasting, we select a diverse set of state-of-the-art baseline models. These include Transformer-based models (iTransformer Liu et al. (2023a), DeformableTST Luo & Wang (2024a), PatchTST Nie et al. (2023), Crossformer Zhang & Yan (2023), and Leddam Yu et al. (2024a)), CNN-based models (TimesNet Wu et al. (2023) and ModernTCN Luo & Wang (2024b)), MLP-based models (DLinear Zeng et al. (2023) and TimeMixer Wang et al. (2024b)), and Fourier-based models (FBM Yang et al. (2024), FITS Xu et al. (2023), and FreMLP Yi et al. (2023)). For short-term forecasting, we further include three strong baseline models: DUET Qiu et al. (2025), SOFTS Han et al. (2024), and TiDE Das et al. (2023).

**Settings** All experiments are conducted using PyTorch 2.5 with Python 3.10 on an NVIDIA H100 80GB GPU. We employ the Adam optimizer with L1 loss for training. Following prior works Zhou et al. (2021); Wu et al. (2021), we adopt Mean Squared Error (MSE) and Mean Absolute Error (MAE) as evaluation metrics.

**Long-term Forecasting** We conduct long-term forecasting experiments on eight widely-used real-world datasets, including the ETT dataset with its four subsets (ETTh1, ETTh2, ETTm1, ETTm2) Wu et al. (2021), as well as Weather, Electricity, Traffic Wu et al. (2023), and Solar Liu et al. (2023a).

We compare our model with other three Fourier-based models. The lookback window is set to 336 for all baseline models. As shown in Tab. 1, TiF outperforms all baseline models

Table 1: **Average results across seven long-term forecasting datasets compared with other three Fourier-based models.** Best results in **bold**, second-best underlined.

| Model | TiF (Ours) | | FBM | | FITS | | FreMLP | |
|---|---|---|---|---|---|---|---|---|
| Metric | MSE | MAE | MSE | MAE | MSE | MAE | MSE | MAE |
| ETTm1 | **0.347** | **0.369** | 0.348 | 0.378 | 0.361 | 0.379 | 0.380 | 0.404 |
| ETTm2 | **0.251** | **0.305** | 0.258 | 0.317 | 0.258 | 0.315 | 0.303 | 0.354 |
| ETTh1 | **0.399** | **0.418** | 0.414 | 0.427 | 0.406 | 0.424 | 0.480 | 0.479 |
| ETTh2 | **0.320** | **0.368** | 0.347 | 0.389 | 0.327 | 0.378 | 0.495 | 0.493 |
| Weather | **0.222** | **0.255** | 0.227 | 0.264 | 0.228 | 0.266 | 0.234 | 0.282 |
| Electricity | **0.156** | **0.251** | 0.164 | 0.257 | 0.173 | 0.266 | 0.175 | 0.273 |
| Traffic | **0.382** | **0.236** | 0.412 | 0.278 | 0.446 | 0.293 | 0.483 | 0.321 |

on both MAE and MSE evaluation metrics. This improvement stems from our innovative backbone and architecture design, achieving superior representation learning capability compared to existing Fourier-based models.

We compare our model with other state-of-the-art baseline models. In all cases, we employ grid search to determine the optimal lookback window from the set $\{96, 192, 336, 512, 720\}$ and other hyperparameters. As shown in Tab. 2, TiF consistently achieves the best model performance. Notably, for the large-scale Traffic dataset, with its 862 variables, poses substantial challenges due to its complex spatiotemporal dependencies. TiF effectively balances within-variable and cross-variable modeling, achieving consistently strong forecasting performance. This robust performance of our model across diverse datasets provides compelling evidence of its practicality and generalizability in the real-world settings.

Table 2: **Average results for all models across eight long-term forecasting datasets.** This table provides a summary of the full results presented in the Appendix. Best results are **bold** on a pale gold background; second-best are underlined on a light green background. Lower values indicate better performance.

| Dataset | TIF (Ours) | | iTransformer (2023a) | | DeformableTST (2024a) | | TimeMixer (2024b) | | PatchTST (2023) | | Crossformer (2023) | | Leddam (2024a) | | ModernTCN (2024b) | | TimesNet (2023) | | DLinear (2023) | |
|---|---|---|---|---|---|---|---|---|---|---|---|---|---|---|---|---|---|---|---|---|
| | MSE | MAE | MSE | MAE | MSE | MAE | MSE | MAE | MSE | MAE | MSE | MAE | MSE | MAE | MSE | MAE | MSE | MAE | MSE | MAE |
| ETTm1 | **0.341** | **0.367** | 0.362 | 0.391 | 0.348 | 0.383 | 0.355 | 0.380 | 0.353 | 0.382 | 0.420 | 0.435 | 0.354 | 0.381 | 0.351 | 0.381 | 0.400 | 0.406 | 0.357 | 0.379 |
| ETTm2 | **0.251** | **0.306** | 0.269 | 0.329 | 0.257 | 0.319 | 0.257 | 0.318 | 0.256 | 0.317 | 0.518 | 0.501 | 0.265 | 0.320 | 0.253 | 0.314 | 0.291 | 0.333 | 0.267 | 0.332 |
| ETTh1 | **0.393** | **0.417** | 0.439 | 0.448 | 0.404 | 0.423 | 0.427 | 0.441 | 0.413 | 0.434 | 0.440 | 0.463 | 0.415 | 0.430 | 0.404 | 0.420 | 0.458 | 0.450 | 0.423 | 0.437 |
| ETTh2 | **0.316** | **0.368** | 0.374 | 0.406 | 0.328 | 0.377 | 0.349 | 0.397 | 0.324 | 0.381 | 0.809 | 0.658 | 0.345 | 0.391 | 0.322 | 0.379 | 0.414 | 0.427 | 0.431 | 0.447 |
| Weather | **0.220** | **0.254** | 0.233 | 0.271 | 0.221 | 0.262 | 0.226 | 0.264 | 0.226 | 0.264 | 0.228 | 0.287 | 0.226 | 0.264 | 0.224 | 0.264 | 0.259 | 0.287 | 0.240 | 0.300 |
| Electricity | **0.152** | **0.248** | 0.164 | 0.261 | 0.161 | 0.261 | 0.185 | 0.284 | 0.159 | 0.253 | 0.181 | 0.279 | 0.162 | 0.256 | 0.156 | 0.253 | 0.192 | 0.295 | 0.166 | 0.264 |
| Traffic | **0.370** | **0.233** | 0.397 | 0.282 | 0.391 | 0.278 | 0.409 | 0.279 | 0.391 | 0.264 | 0.523 | 0.284 | 0.452 | 0.283 | 0.396 | 0.270 | 0.620 | 0.336 | 0.434 | 0.295 |
| Solar | **0.182** | **0.213** | 0.200 | 0.260 | 0.185 | 0.254 | 0.193 | 0.252 | 0.194 | 0.245 | 0.191 | 0.242 | 0.223 | 0.264 | 0.228 | 0.282 | 0.244 | 0.334 | 0.247 | 0.309 |

**Short-term Forecasting** For short-term forecasting, we conduct experiments on the PeMS datasets Liu et al. (2022), which the complex spatiotemporal dependencies of urban transportation networks. The lookback window is set to 96 for all baseline models.

Some models, including PatchTST Nie et al. (2023) and DLinear Zeng et al. (2023), achieve strong performance in long-term forecasting using channel-independent technique. However, these models show significant performance drops on the PeMS dataset. This because this dataset has strong cross-variable correlations. In contrast, our model maintains robust performance on this challenging task. As shown in Tab. 3, TiF consistently outperforms all baseline models, demonstrating its effectiveness in capturing complex spatiotemporal dependencies.

Table 3: **Average results for all models across four short-term forecasting datasets.** This table provides a summary of the full results presented in the Appendix. Best results are **bold** on a pale gold background; second-best are underlined on a light green background. Lower values indicate better performance.

| Model | TiF (Ours) | | DUET (2025) | | iTransformer (2023a) | | Leddam (2024a) | | SOFTS (2024) | | PatchTST (2023) | | Crossformer (2023) | | TimesNet (2023) | | TiDE (2023) | | DLinear (2023) | |
|---|---|---|---|---|---|---|---|---|---|---|---|---|---|---|---|---|---|---|---|---|---|
| | MSE | MAE | MSE | MAE | MSE | MAE | MSE | MAE | MSE | MAE | MSE | MAE | MSE | MAE | MSE | MAE | MSE | MAE | MSE | MAE |
| PeMS03 | **0.082** | **0.186** | 0.086 | 0.192 | 0.096 | 0.204 | 0.101 | 0.210 | 0.087 | 0.192 | 0.151 | 0.265 | 0.138 | 0.253 | 0.119 | 0.271 | 0.271 | 0.380 | 0.219 | 0.295 |
| PeMS04 | **0.087** | **0.190** | 0.096 | 0.203 | 0.098 | 0.207 | 0.102 | 0.213 | 0.091 | 0.196 | 0.162 | 0.273 | 0.145 | 0.267 | 0.109 | 0.220 | 0.307 | 0.405 | 0.236 | 0.350 |
| PeMS07 | **0.069** | **0.161** | 0.076 | 0.176 | 0.088 | 0.190 | 0.087 | 0.192 | 0.075 | 0.173 | 0.166 | 0.270 | 0.181 | 0.272 | 0.106 | 0.208 | 0.297 | 0.394 | 0.241 | 0.343 |
| PeMS08 | **0.091** | **0.186** | 0.096 | 0.192 | 0.127 | 0.212 | 0.102 | 0.211 | 0.114 | 0.208 | 0.238 | 0.289 | 0.232 | 0.270 | 0.150 | 0.244 | 0.347 | 0.421 | 0.281 | 0.366 |

Table 4: **Ablation study.** Best results in bold, second-best underlined.

| Model | TiF | | w/o AT | | w/o SA | |
|---|---|---|---|---|---|---|
| Metric | MSE | MAE | MSE | MAE | MSE | MAE |
| ETTm1 | **0.341** | **0.367** | 0.347 | 0.369 | 0.354 | 0.376 |
| ETTh1 | **0.393** | **0.417** | 0.402 | 0.422 | 0.403 | 0.423 |

| Model | TiF | | w/o LS | | w/o GS | |
|---|---|---|---|---|---|---|
| Metric | MSE | MAE | MSE | MAE | MSE | MAE |
| PeMS03 | **0.082** | **0.186** | 0.094 | 0.199 | 0.087 | 0.193 |
| PeMS08 | **0.091** | **0.186** | 0.101 | 0.197 | 0.098 | 0.194 |

**Ablation Study** To validate the effectiveness of our proposed backbone and architecture, we conduct a comprehensive ablation study on representative datasets, as shown in Tab. 4. Specifically, we evaluate the impact of Adaptive Truncation (AT) and Selective Activation (SA) on the ETTm1 and ETTh1 datasets. The experimental results demonstrate that both components are essential for achieving optimal performance. Subsequently, we assess the contributions of Local Spatial (LS) and Global Spatial (GS) information on the PeMS03 and PeMS08 datasets. The experimental results confirms that both local and global spatial modeling are crucial for effective spatiotemporal modeling.

**Efficiency Analysis** We comprehensively compare the forecasting performance, training speed, and memory consumption of our model against other baseline models, where lookback window and forecast horizon are both set to 96. As shown in Fig. 4, TiF outperforms other Transformer-based and CNN-based models in terms of model efficiency.

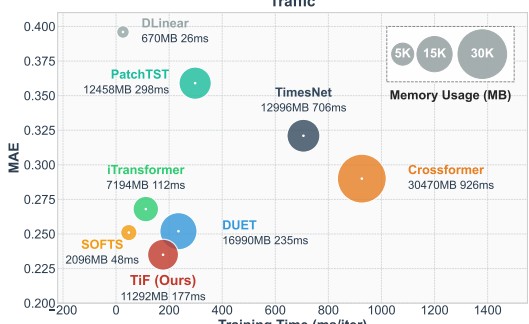

Figure 4: **Model efficiency comparison on the Traffic dataset in terms of MAE and training time.**

## 6 CONCLUSION

**Limitations** While our model achieves state-of-the-art performance in multivariate time series forecasting, three key limitations remain: (1) reduced effectiveness on highly irregular or event-driven sequences, and (2) untested performance on other time series tasks such as imputation, classification, and anomaly detection.

**Broader Impact** Our work delivers tangible benefits, including better forecasting performance and potential improvement in resource allocation across various domains. However, potential risks include perpetuation of historical information biases , and over-reliance on automated forecasting in the absence of adequate human oversight. We encourage ongoing research to address these concerns while maximizing the positive impact of advanced forecasting capability.

## 7 ETHICS STATEMENT

This work does not involve human subjects, does not raise concerns regarding data privacy, bias, fairness, or potential harmful applications, and does not present conflicts of interest or legal compliance issues. The research methodology and findings do not pose ethical concerns that require additional consideration beyond standard academic practices.

## 8 REPRODUCIBILITY STATEMENT

To ensure reproducibility of our results, we provide the following resources: (1) complete implementation details and hyperparameters are described in Section 5 and Appendix C; (2) all datasets used in our experiments are publicly available and properly cited with access information provided in Section 5 and Appendix B; (3) theoretical proofs and derivations are included in Section 3; and (4) source code will be made available upon publication to facilitate replication of our experimental results.

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
