# OpenReview forum: "Multivariate Time Series Forecasting with Fourier Neural Filter"
_ICLR.cc/2026/Conference — ICLR 2026 Conference Withdrawn Submission_

### Official Review · Reviewer_oAZB · 2025-10-16

**Soundness:** 2
**Presentation:** 2
**Contribution:** 2
**Rating:** 4
**Confidence:** 4

**Summary:**

The paper proposes Time Filter (TiF), combining a new Fourier Neural Filter (FNF) backbone with a Dual Branch Decoupler (DBD) architecture. FNF extends FNO with an input-dependent kernel (selective activation), complex transform, and adaptive truncation to modulate time–frequency information; DBD uses parallel temporal/spatial branches to improve gradient flow and capacity from an information-bottleneck perspective. Experiments on 12 benchmarks (eight LTSF datasets plus four PeMS short-term sets) report strong results, along with ablations and an efficiency plot.

**Strengths:**

1. FNF formalizes an input-dependent spectral operator with selective activation and adaptive truncation; the math exposition (Definitions. 1–7, Remarks 1–6) is explicit and links capabilities to Transformer functions and complexity.

2. DBD’s motivation via the information bottleneck and gradient-path analysis is well argued.

3. Results span LTSF and PeMS with comparisons to Transformer/CNN/MLP/Fourier baselines; Table 2 claims lookback grid search (96–720) for all methods, which ensures a fair comparison.

4. Efficiency analysis (Traffic) and component ablations (AT/SA, LS/GS) give some insight.

**Weaknesses:**

1. FNF’s contributions (input-dependent kernel, selective activation, adaptive truncation) are close in spirit to prior spectral/fractional operators (e.g., FNO/AFNO/FITS/FreMLP). The paper does not clearly establish what capability FNF enables that prior spectral blocks cannot, beyond architectural composition; DBD overlaps with known parallel/dual-path decouplers (e.g., Leddam [1], xPatch [2], MTGNN [3], TimeMixer++ [4], just to name a few). A sharper comparison or operator-replacement study is needed.

2. There is no working anonymous repo or pseudo code for review, which limits the reproducibility of the proposed method.

3. For PeMS, the lookback is fixed to 96 for all baselines, which can bias results; fair practice tunes input length per method (as you already did for Table 2). Please align protocols across tasks.

4. Table 2 lacks more recent strong baselines (e.g., TimeMixer++ [4] (ICLR25), PatchMLP [5] (AAAI25), TQNet [6], and TimeBridge [7] (ICML25)), which undermines the SOTA claim; please add with identical splits/tuning.

5. The proposed FNF looks conceptually overlapped with selective state-space models such as Mamba: both realize an input-conditioned long filter with a gated residual/skip path. In FNF, the frequency-domain parameterization (F → complex transform → adaptive truncation → F⁻¹) effectively implements a learnable long convolution; Mamba parameterizes a similar operation via SSM kernels and a selective gate. The manuscript should explicitly position FNF against Mamba/S4-family—clarifying what FNF does that a selective SSM cannot—and include quantitative comparisons.

*[1] Revitalizing Multivariate Time Series Forecasting: Learnable Decomposition with Inter-Series Dependencies and Intra-Series Variations Modeling.*

*[2] xPatch: Dual-Stream Time Series Forecasting with Exponential Seasonal-Trend Decomposition.*

*[3] Connecting the Dots: Multivariate Time Series Forecasting with Graph Neural Networks.*

*[4] TimeMixer++: A General Time Series Pattern Machine for Universal Predictive Analysis.*

*[5] Unlocking the Power of Patch: Patch-Based MLP for Long-Term Time Series Forecasting.*

*[6] Temporal Query Network for Efficient Multivariate Time Series Forecasting.*

*[7] TimeBridge: Non-Stationarity Matters for Long-term Time Series Forecasting.*

**Questions:**

See in weakness.

---

### Official Review · Reviewer_RTz3 · 2025-10-26

**Soundness:** 2
**Presentation:** 3
**Contribution:** 2
**Rating:** 4
**Confidence:** 4

**Summary:**

This paper introduces a new architecture for multivariate time series forecasting, including two key components:
(1) Fourier Neural Filter (FNF), an input-dependent integral kernel operator that unifies time-domain and frequency-domain modeling, extending the Fourier Neural Operator (FNO) by introducing adaptive gating, selective activation, and learnable truncation for denoising.
(2) Dual Branch Decoupler (DBD), a dual-path structure inspired by information-bottleneck theory that decouples temporal and spatial processing for improved gradient flow and representation capacity.

**Strengths:**

1. This paper presents a new architectural exploration for time series forecasting, offering a meaningful attempt to design a dedicated backbone tailored to the characteristics of temporal data. This represents a positive and constructive step for research in this area.

2. The proposed Dual Branch Decoupler (DBD) introduces a parallel-branch mechanism to decouple temporal and spatial feature learning. This is an interesting design that contributes fresh insights to spatiotemporal modeling in time series forecasting.

3. The experimental evaluation is extensive and convincing, covering 12 benchmark datasets and a broad spectrum of competitive baseline models, which demonstrates the robustness and general applicability of the proposed approach.

**Weaknesses:**

1. While the proposed Fourier Neural Filter (FNF) introduces adaptive kernels and learnable truncation mechanisms, much of its formulation builds upon existing frameworks such as FNO and AFNO.

2. The DBD parallel design is conceptually sound but lacks empirical exploration of branch interactions (e.g., information flow visualization or mutual information analysis). It would strengthen the paper to show why the parallel path quantitatively improves gradient dynamics or representation diversity.

3. The Related Work section does not sufficiently discuss prior studies directly related to the paper’s two main contributions, i.e., FNF and DBD.

4. The ablation study is relatively limited. It would be useful to further investigate the effect of architectural choices, such as FNF depth, kernel size, and sensitivity to the patch length P, to better understand the robustness of the proposed design.

**Questions:**

1. It is unclear why Equation (22) is claimed to capture global correlations while Equation (21) captures local correlations, given that both modules employ the Fourier Neural Filter (FNF) backbone.

---

### Official Review · Reviewer_9JoG · 2025-10-31

**Soundness:** 2
**Presentation:** 2
**Contribution:** 2
**Rating:** 4
**Confidence:** 4

**Summary:**

This paper proposes a Time Filter (TiF) for multivariate time-series forecasting. TiF employs a Fourier Neural Filter (FNF) as the backbone and a Dual-Branch Decoupler (DBD) as the architectural design. The former provides strong representational capacity, while the latter establishes efficient learning pathways for spatiotemporal modeling.

**Strengths:**

1. This paper proposes a unified FNF backbone that integrates time-domain and frequency-domain analyses.
2. This paper provides theoretical and empirical evidence for the effectiveness of DBD in spatiotemporal modeling.
3. Comprehensive experiments conducted on long-term and short-term forecasting tasks verify the superior performance of TiF.

**Weaknesses:**

1. The organization needs improvement to make it easier to follow. In the Related Work, it is unclear why distribution shift and non-autoregressive decoding are reviewed, as these topics do not appear to be central to the paper’s main contributions. In the Method (Sections 3.1.1–3.1.6), substantial space is devoted to preliminaries such as complex transforms and global convolution, which obscures the core ideas and innovations of the proposed approach.
2. The paper’s novelty appears limited. Simply replacing the fixed kernel in the Fourier Neural Operator with an input-dependent kernel represents only a marginal improvement. In addition, introducing DBD as a parallel paradigm, compared with unified and sequential paradigms, to maintain independent information-processing branches also appears to be an incremental design choice rather than a substantive conceptual advance.
3. More relevant frequency-filter baselines should be considered, such as FilterNet [1] and TSLANet [2].

[1] FilterNet: Harnessing Frequency Filters for Time Series Forecasting. NeurIPS, 2024.
[2] TSLANet: Rethinking Transformers for Time Series Representation Learning. ICML, 2024.

**Questions:**

pls refer to the weakness.

---

### Official Review · Reviewer_K5pj · 2025-11-01

**Soundness:** 3
**Presentation:** 1
**Contribution:** 2
**Rating:** 4
**Confidence:** 3

**Summary:**

This paper introduces Time Filter (TiF) structure for multivariate time series forecasting. TiF combines: 1) Fourier Neural Filter (FNF), a spectral backbone with an input-dependent kernel that adaptively mixes time-domain and frequency-domain information while filtering noise; 2) Dual Branch Decoupler (DBD), a parallel temporal–spatial architecture that processes time and variable dimensions separately and fuses them later. The paper provides theoretical motivation via the information bottleneck principle and demonstrates improved performance across 12 datasets with notable efficiency and robustness.

**Strengths:**

1. The method design is well-motivated. The proposed FNF introduces an adaptive, input-dependent spectral filter that effectively bridges time-domain and frequency-domain modeling, while the DBD offers a parallel architecture for capturing both temporal and spatial dependencies.

2. The paper is written clearly, with good method framing and empirical support, making it a sounding backbones for time series forecasting.

3. Consistent better results across 12 datasets demonstrates the generalization and efficiency advantages of the proposed method.

**Weaknesses:**

1. The theoretical claims in this paper are mostly definitions or intuition. Theoretical proofs in Sections 3.1 and 3.2 read more as descriptive derivations or qualitative reasoning rather than rigorous theorems or guarantees. To strengthen credibility, the authors could either (a) provide concrete theoretical statements with clear conditions and supporting lemmas, or (b) reframe these sections as design intuitions supported by experiments.

2. The experiment section in the main paper lacks sufficient description of hyperparameters and training details. The authors state that a grid search over input lookback lengths and other hyperparameters was performed. However, it is unclear how this grid search was conducted. Is the performance of the proposed method significantly affected by the choice of lookback window length? Additional experiments using a fixed window size and including statistical significance tests would help address these concerns.

3. The length distribution across sections could be better balanced. Sections 3 and 4 occupy a large portion of the paper, leaving the experimental section relatively brief. If Section 3 mainly focuses on describing the model architecture, it could be condensed to make room for more detailed experiments. Similarly, Section 4 could be merged into Section 3. At present, the ablation studies are rather limited, discussing only a few modules and not conducted on unified dataset selection.

**Questions:**

1. The paper claims selective activation enhances mid/high frequencies. Some frequency or distribution visualizations would make this more concrete.

2. Providing pseudo-code for the structures or releasing an anonymized codebase with experimental configs would benifit transparency and reproducibility.

3. How do you design the fusion network for the representations? What is the dimension of the linear layer in Eq. 23? Why it is a Linear - LN - GELU - Linear structure, which is not a common style MLP?

If the authors' response adequately addresses my questions and concerns mentioned above, I am willing to raise my score.

---

### Note · Authors · 2026-01-26

I have read and agree with the venue's withdrawal policy on behalf of myself and my co-authors.

---

### Meta-Review · Area_Chair_LCsz · 2026-01-16

**Summary:**

This paper proposes a new method for spatiotemporal modeling with a component that "integrates time-domain and frequency-domain analysis" and an architecture that provides "optimal learning pathways" leveraging new theoretical findings from the perspective of the information bottleneck. The evaluation is performed on a set of 12 public datasets, comprising "multivariate long-term and short-term forecasting tasks".


I based my assessment on the following concerns, derived from the reviews:

W1. The theoretical results in this paper were seen as unrigorous.

W2. The novelty of the resulting method was seen as limited, with one reviewer noting the contribution as "simply replacing the fixed kernel in the Fourier Neural Operator with an input-dependent kernel".

W3. No code was provided, raising questions of reproducibility.

W4. The experiments lacked recent baselines.

**Reviewer Concerns:**

There was no author response.


Considering there was no response, I agree with the reviewers' assessment that the paper, while presenting a well-motivated method with some features of interest in using the information bottleneck and gradient flow, is not ready for publication due to its limited novelty, unimpressive theoretical results and deficiencies in including recent baselines.

**Reviewer Scores:**

The reviewers would have likely not changed their scores, as there was no response from the authors.

---

### Decision · Program_Chairs · 2026-01-26

Reject